# Live Biotherapeutic *Lactococcus lactis* GEN3013 Enhances Antitumor Efficacy of Cancer Treatment via Modulation of Cancer Progression and Immune System

**DOI:** 10.3390/cancers14174083

**Published:** 2022-08-23

**Authors:** Sujeong Kim, Yunjae Kim, Suro Lee, Yulha Kim, Byungkwan Jeon, Hyerim Kim, Hansoo Park

**Affiliations:** 1Department of Biomedical Science and Engineering, Gwangju Institute of Science and Technology (GIST), Gwangju 61005, Korea; 2Genome and Company, Pangyo-ro 255, Bundang-gu, Seoungnam 13486, Korea

**Keywords:** *Lactococcus lactis*, tumor growth, immune system, chemotherapy, immunotherapy

## Abstract

**Simple Summary:**

Recent studies, which have revealed the strong relationship between gut microbiota and tumor progression, have driven the clinical application of microbiome-based treatments to increase the efficacy of anticancer therapies. In particular, the genome-editing *Lactococcus lactis*, which activates the host immune system by expressing immune-boosting cytokines or metabolites, is a candidate for microbiome treatment. While *Lactococcus lactis* has so far been studied in terms of its recombinant forms, we investigated the anticancer effects of the strain-specific *Lactococcus lactis* GEN3013 itself. In vitro cytotoxicity tests showed that *L. lactis* GEN3013 inhibited the cell growth of various human and murine cancer cell lines. Consistent with the in vitro results, *L. lactis* GEN3013 showed antitumor effects and enhanced the therapeutic efficacy of both chemotherapy and immunotherapy in syngeneic mice. In addition, the host immune system was activated both locally and systemically by the combinatorial treatment of *L. lactis* GEN3013 with chemotherapy and immunotherapy. For these reasons, we suggest that *L. lactis* GEN3013 could be utilized as a novel biotherapeutic agent for cancer treatment.

**Abstract:**

The gut microbiota is responsible for differential anticancer drug efficacies by modulating the host immune system and the tumor microenvironment. Interestingly, this differential effect is highly strain-specific. For example, certain strains can directly suppress tumor growth and enhance antitumor immunity; however, others do not have such an effect or even promote tumor growth. Identifying effective strains that possess antitumor effects is key for developing live biotherapeutic anticancer products. Here, we found that *Lactococcus lactis* GEN3013 inhibits tumor growth by regulating tumor angiogenesis and directly inducing cancer cell death. Moreover, *L. lactis* GEN3013 enhanced the therapeutic effects of oxaliplatin and the PD-1 blockade. Comprehensive immune profiling showed that *L. lactis* GEN3013 augmented cytotoxic immune cell populations, such as CD4^+^ T cells, CD8^+^ effector T cells, and NK cells in the tumor microenvironment. Our results indicate that *L. lactis* GEN3013 is a promising candidate for potentiating cancer treatment in combination with current standard therapy.

## 1. Introduction

A gut microbiome and a host have a symbiotic relationship, and the gut microbiome maintains host health homeostasis [1,2,3]. Therefore, research on the association between the gut microbiome and various diseases is ongoing. In particular, the gut microbiome affects not only the gut’s immune system, but also the host’s immune system in general. For example, orally administered *Bifidobacterium spp.* increases the number of activated T lymphocytes in the intestines and spleen and reduces tumor-associated inflammation [4,5]. Additionally, metabolites produced by the microbiome, such as toxins, PAMPs, and short-chain fatty acids, can suppress tumor growth by having cytotoxic effects on rapidly replicating cancer cells or by inhibiting angiogenesis [6,7]. Given this evidence, the microbiome is an emerging modality for treating cancer, in combination with chemotherapy or immunotherapy [8,9,10,11,12,13,14]. Some studies have suggested that cancer chemotherapy efficacy and outcomes are associated with the gut microbiome. For instance, strain-specific *Bifidobacterium breve* improved the efficacy of oxaliplatin in a syngeneic mouse model, and targeting *F. nucleatum* in the tumor augmented the effect of chemotherapy against colorectal cancers. Other studies have shown that effective microbiota can influence the therapeutic response to immunotherapy [9,15,16]. In a study, it was found that when mice that were non-responsive to anti-PD-L1 treatment were orally given feces isolated from responding mice or *Bifidobacterium* alone, their response to the same therapy increased [17]. Another study revealed a positive correlation between the immunotherapy response and the relative abundance of *Akkermansia muciniphila*. Furthermore, supplementation with *A. muciniphila* restored the response to the PD-1 blockade through the stimulation of specific T lymphocyte recruitment into the tumor microenvironment, even for FMT with non-responder feces [14].

*Lactococcus lactis*, a lactic acid-producing bacterium, is known for its safety and efficiency in genetic engineering. Therefore, recent studies have introduced recombinant *L. lactis* as a live vaccine bacterium for delivering antigens to the host immune system [18]. TRAIL-expressing or IL-12-secreting recombinant *L. lactis* has been shown to induce the apoptosis of cancer cells or an immune response [19,20,21,22]. *L. lactis* itself, however, has not been considered as a live biotherapeutic product, although some studies have revealed that it inhibits the proliferation and induces the apoptosis of cancer cells in stomach and colorectal cancers [23,24,25,26].

To develop a novel anticancer therapy or improve the therapeutic efficacy of conventional approaches, immune cells such as CD4^+^ T cells, CD8^+^ cytotoxic T cells, and NK cells are critical. In particular, the immune cells must be infiltrated and activated in the tumor microenvironment. CD4^+^ T cells mediate antitumor immunity by assisting CD8^+^ cytotoxic T cells and triggering antibody responses [27]. CD8^+^ cytotoxic T cells play a role in directly killing cancer cells, and cancer treatment is only successful if these cells are activated [28]. These T cells exhibit antitumor immunity through the secretion of effector cytokines, such as IFN-γ [29], TNF-α [30], and interleukin-2 [31], in the tumor microenvironment. NK cells are also key immune effector cells, capable of directly killing cancer cells without MHC specificity and of triggering adaptive immune responses [32,33,34].

For these reasons, identifying effective microbiomes with antitumor effects is an important step in developing novel anticancer modalities and enhancing conventional anticancer therapies. Here, we identified that a specific *L. lactis* GEN3013 strain isolated from a healthy individual had antitumor immunity and cytotoxic effects against cancer. *L. lactis* GEN3013 inhibited the growth of various cancer cell lines in vitro and induced tumor growth retardation in a syngeneic mice model. Moreover, it enhanced the efficacy of chemotherapy and immunotherapy by increasing the number of NK and T cells in both the spleen and tumor. These results indicate that *L. lactis* GEN3013 could be used as a live biotherapeutic product to benefit cancer treatment.

## 2. Materials and Methods

### 2.1. Mice 

All animal experiments were approved by the Institutional Animal Care and Use Committee of CHA University and KLSbio. The mice used here were maintained and handled according to policies approved by CHA University and KLSbio, respectively. Five-week-old female C57BL/6 mice were purchased from Orient Bio (Gyeonggi, Korea). For the syngeneic tumor mouse model, a total of 2 × 10^5^ MC38 cells were subcutaneously inoculated in C57BL/6 mice. The mice were orally administered the *L. lactis* strain daily and were intra-peritoneally treated with anti-PD-1 mAb (clone RMP1-14; BioXCell, Lebanon, NH, USA) or 3 mg/kg oxaliplatin (S1224, Selleckchem, Houston, TX, USA) on days 1, 4, 8, 11, and 14. The tumor size was monitored three times per week until the study endpoint. The tumor volume was calculated as the length × width^2^ × 0.5.

Nude mice were purchased from Orient Bio (Gyeonggi, Korea). Nude mice were orally administered the *L. lactis* strain for 14 days. Then, a total of 2 × 10^5^ CT26, 1 × 10^6^ 4T1, and 1 × 10^6^ HCT116 cells were inoculated subcutaneously into the flank of each mouse. The nude tumor mice were administered the *L. lactis* daily, and the tumor size was monitored three times a week until the study endpoint. The tumor volume was calculated as the length × width^2^ × 0.5.

### 2.2. Cell Culture 

Human lung (H1975, H1299, A549, and HCC827), colon (LoVo and HCT116), gastric (SNU-601, MKN-1, SNU-216, and AGS), breast (MDA-MB-231, Hs578T, and BT20), and liver (HepG2) cancer cell lines were purchased from the Korean Cell Line Bank (Jongno-gu, Seoul, Korea). Mouse lung (LLC1), colon (CT26 and MC38), and breast (4T1) cancer cells were purchased from the American Type Culture Collection (Manassas, VA, USA) and Kerafast (Winston-Salem, NC, USA). All cells were free from mycoplasma contamination. All human cell lines were cultured in Roswell Park Memorial Institute medium (RPMI, GIBCO) and all mouse cell lines were cultured in Dulbecco’s modified eagle medium (DMEM, GIBCO). Each medium was supplemented with 10% fetal bovine serum (FBS, GIBCO), 100 units/mL penicillin (GIBCO), and streptomycin (GIBCO). All cells were maintained in a 5% CO_2_ humidified incubator at 37 °C.

### 2.3. Bacteria 

*L. lactis* GEN3013 was isolated from a healthy donor. Stool samples were collected from healthy donors from the Gwangju Institute of Science and Technology with informed consent. This study was approved by the institutional review board (20192009-BR-48-02-04). Fresh fecal samples (approximately 4.0 g) were transferred into bottles containing 80 mL of PBS and resuspended by vortexing in an anaerobic chamber. The resulting fecal fluids were serially diluted 10^−4^–10^−8^-fold with PBS and 100 or 200 μL of these was spread onto various agar media: MRS (pH 6.8, 5.2, and 3.5), MRS-cysteine, TOS-propionate, brain heart infusion, Colombian base blood, Eggerth–Gagnon, Gifu anaerobic medium (GAM), M2GSC, modified YCFA, mucin, and reinforced clostridial medium. After inoculation, the plates were placed in the bacteria incubator under aerobic conditions at 37 °C for 48–72 h. Individual colonies from the incubated plates were picked and prepared to sequence the 16S rRNA regions. Colony PCR was performed using the following parameters: pre-denaturation at 95 °C for 15 min, denaturation at 95 °C for 30 s, annealing at 55 °C for 30 s, extension at 72 °C for 1 min and 45 s, and a final extension at 72 °C for 5 min. The denaturation, annealing, and extension steps were repeated 32 times. The presence of PCR bands was confirmed via agarose gel (1%) analysis, and the PCR sample was purified using a PCR purification kit. The universal primers 27F and 1492R (V1-V9 region) and 518F and 800R (V4 region) were used (Macrogen Co., Seoul, Korea). The obtained nucleotide sequences were trimmed to remove the unclear nucleotides and were assembled (5′ forward and 3′ reverse orientation). Then, the modified sequences were placed in the NCBI BLAST program to identify the phylogenetically closest 16S rRNA gene sequences.

### 2.4. Bacterial Sampling

For obtaining bacterial lysates, bacterial cells were collected via centrifugation at 7000× *g* for 5 min at 4 °C. In order to remove the bacterial culture media, bacterial cells were washed twice with PBS. The bacterial cells were re-suspended in PBS until the 600 nm optical density (OD600) was 100, and were then placed in a 2 mL lysing matrix tube (MPBio, Santa Ana, CA, USA). By using the bead-beating method with Fast-Prep-24 5G (MPBio), the bacterial cells were homogenized. The bead-beating conditions were as follows: the speed was 6.0 m/s, the beating time was 30 s, the pause time was 300 s, and this procedure was repeated three times. The bacterial lysates were centrifuged at 12,000 rpm for 10 min at 4 °C. After serially filtering the supernatants with 0.45 µm and 0.2 µm filters, they were subsequently transferred to new tubes and stored at −80 °C until further assay.

Conditioned bacterial supernatants were obtained after the bacteria had adapted to the cell culture media (RPMI or DMEM). In detail, when a bacterial culture reached the appropriate number of CFUs, the bacterial culture media was mixed with cell culture media in a ratio of 8:1, and the mixture was incubated at 37 °C overnight. The cell culture media-incubated bacteria were centrifuged at 7000× *g* for 5 min at 4 °C, and the supernatant was subsequently transferred to a new tube and stored at −80 °C until further assay.

### 2.5. Metabolic Profiling

Bacterial cells (10^10^ CFUs) were collected via centrifugation at 7000× *g* for 5 min at 4 °C. After freezing the bacterial cells at −80 °C, 200 μL of 80% methanol in water was added to each bacterial cell sample. After vortexing for 1 min at room temperature, the suspension was frozen at −80 °C for 3 h. This extraction procedure was repeated three times. The extracted samples were thawed on ice and immediately centrifuged (10,000× *g*) for 2 min at 4 °C. The supernatant was subsequently transferred to a new tube and stored at −80 °C until further analysis.

For the untargeted analysis, 10 μL of sulfamethoxazole (2200 μg/L) was added to 100 μL of the extracted sample as an internal standard for normalization. The supernatants were transferred to sample vials and analyzed using UHPLC-ESI-Q-TOF/MS (UHPLC: DIONEX UltiMate 3000, Dionex Corporation, Sunnyvale, CA, USA; MS: TOF 5600+, AB Sciex, Path Framingham, MA, USA). UHPLC separation was performed using a CORTECS C18 column (2.1 mm × 100 mm, 1.6 μm; Waters). Mobile phase A consisted of 5 mM ammonium acetate in water containing 0.1% acetic acid (*v*/*v*), whereas mobile phase B contained 0.05% formic acid (*v*/*v*) in acetonitrile. Samples were eluted using the following conditions: 5% of mobile phase B increased to 95% at 35 min and equilibrated for an additional 5 min. The flow rate used was 0.4 mL/min. The column temperature was maintained at 45 °C. Mass acquisition was performed in both positive (ESI+) and negative (ESI−) electrospray ionization modes using the following parameters: a collision energy of 35 ± 10 eV; a capillary voltage of 13,800 V (positive mode) and 23,800 V (negative mode), with nozzle voltages of 10 V and 20 V, respectively; a gas temperature of 300 °C; a drying gas (nitrogen) rate of 7 l/min; a nebulizer gas (nitrogen) pressure of 40 psi; a sheath gas temperature of 330 °C; and a flow rate of 10 l/min. Mass data were collected in the 80–1700 m/z range at an acquisition rate of 2 spectra/s. The injection volume used was 3 μL.

For metabolite identification, the following resources and parameters were used: the metabolites detected in each group were analyzed using Scaffold Elements v. 2.1.1 (Proteome Software, USA) for the spectrum properties and filtering (S/N > 3); peak alignment (RT ≤ 0.5 min and MT ≤ 4 ppm), feature extraction (adduct ions: [M + H]^+^, [M + NH_4_]^+^, [M + Na]^+^, and [M-nH_2_O + H]^+^ for ESI (+) and [M − H]^−^, [2M − H]^−^, [M + HCOO]^−^, and [M-nH_2_O-H]^−^ for ESI (−); peak with an S/N ≥ 10 and an intensity ≥ 1000), and feature grouping (RT ≤ 0.5 min and MT ≤ 4 ppm); and the MS/MS spectra of the compounds and metabolome databases, including the Human Metabolomics Database (http://www.hmdb.ca/ (accessed on 7 January 2022)), METLIN (http://metlin.scripps.edu/ (accessed on 11 January 2022)), MassBank (http://www.massbank.jp/ (accessed on 19 January 2022)), NIST (http://www.nist.gov/ (accessed on 8 February 2022)), and LIPID MAPS (http://www.lipidmaps.org/ (accessed on 14 February 2022)).

### 2.6. MTT Assay

The cancer cell lines were dispensed in a 96-well plate at 1–5 × 10^3^ cells/well, and after 24 h, the bacterial samples (bacterial lysates or conditioned bacterial supernatants) were added at 0.5% and incubated for 72 h. Next, each well was treated and incubated with 3-(4,5-dimethylthiazol-2-yl)-2,5-diphenyltetrazolium bromide thiazolyl blue (MTT) reagent for 2 h. The mitochondria of living cells converted the yellow MTT to purple formazan. Afterwards, the culture medium containing MTT was removed, and 100 μL of DMSO was added to each well. The purple color intensity was measured at 540 nm using a microplate reader.

### 2.7. Colony Forming Assay

Various cancer cell lines were diluted to a density of 1–2 × 10^3^ in each well and incubated for 24 h at 37 °C in a 5% CO_2_ atmosphere. *L. lactis* GEN3013 was cultured in MRS liquid medium (Difco) containing 0.5% glucose and 0.05% cysteine at 37 °C for 24 h, and the bacteria were disrupted. Bacterial lysates were then added to each well, and the medium was exchanged every 2–3 days. After the cells were incubated for 144 h, they were fixed with 4% formalin for 30 min and washed with PBS twice. Finally, the cells were stained with crystal violet for 5 min and observed using a cell culture microscope (CKX53; Olympus, Shinjuku City, Tokyo).

### 2.8. Wound Healing Assay

HCT116, A549, or MC38 cancer cells were diluted to 6 × 10^5^ cells/well on a 6-well plate. When cell confluency reached 90–95%, a scratch was made on the cell monolayer at regular intervals using a 1000 µL tip. Then, PBS or bacterial samples (lysates or conditioned bacterial supernatants) were used to treat cells for 24 h and cell mobility was observed using a cell culture microscope (Olympus, CKX53). Image processing was performed using the “wound healing size tool” plugin of ImageJ (ver. 1.8.0_172).

### 2.9. T Cell Proliferation Assay

PBMCs were isolated from human blood using Ficoll (GE healthcare, Chicago, IL, USA), and the red blood cells were removed using red blood cell (RBC) lysis buffer (Biolegend, San Diego, CA, USA). Monocytes were isolated from RBC-lysed PBMCs and 5 × 10^3^ cells were placed into each well of a 96-well plate. Then, *L. lactis* GEN3013 was added to each well and mixed gently to react with the monocytes for 2 h. During the reaction between the monocytes and *L. lactis* GEN3013, T cells were isolated from the extra PMBCs using MACS buffer and an LS column (Miltenyi Biotec, Bergisch Gladbach, Germany). The isolated T cells were diluted in RPMI medium to a concentration of 5 × 10^4^ cells per 100 μL and aliquoted into each well. They were cultured together for 48 h at 37 °C in a 5% CO_2_ atmosphere. Finally, the cell culture supernatant was collected and the IFN-γ and IL-10 levels in the supernatant were measured using an ELISA kit (Invitrogen, Waltham, MA, USA).

Splenocytes were isolated from the mouse spleens using 70 µm cell strainers, and red blood cells were removed using RBC lysis buffer. CD4^+^ or CD8^+^ T cells were isolated from the splenocytes using MACS buffer and an LS column (Miltenyi Biotec, Bergisch Gladbach, Germany) and stained with CellTrace CFSE (C34554, Invitrogen). Isolated T cells were diluted using RPMI media to a density of 1–2 × 10^5^ cells per 100 μL and aliquoted into each well. They were incubated with the conditioned bacterial supernatants of RPMI-incubated *L. lactis* GEN3013 culture supernatants for 48 h at 37 °C in a 5% CO_2_ atmosphere. Finally, T cell proliferation was analyzed using flow cytometry.

### 2.10. Immune Cell Profiling 

The experimental mice were sacrificed on day 15 after tumor cell implantation. We performed the dissection of tumor tissues into small pieces and transferred them to RPMI 1640 media (GIBCO) containing collagenase I (GIBCO, 2.5 mg/mL), collagenase II (GIBCO, 1.5 mg/mL), collagenase IV (GIBCO, 1 mg/mL), DNase 1 (Thermo Fisher, Waltham, MA, USA, 50 µg/mL), and hyaluronidase type IV-S (Sigma-Aldrich, Burlington, MA, USA, 0.25 mg/mL). The tumor tissues were incubated at 37 °C for 50 min, followed by filtration using a 70 µm cell strainer (BD Bioscience, San Jose, CA, USA). After homogenizing in RPMI 1640 medium, the spleen tissues were lysed in RBC lysis buffer (eBioscience, San Diego, CA, USA) and filtered using a 70 µm cell strainer. To block the Fc receptor, both the splenocytes and tumor cells were treated with anti-mouse CD16/CD32 (BD Biosciences, San Jose, CA, USA) for 10 min at 4 °C. After the assessment of cell viability and cell number, the antibody staining for surface markers was conducted. Then, fixation and permeabilization buffer (BioLegend, San Diego, CA, USA) solution was added, followed by antibody staining for intracellular markers.

The following antibodies were used to stain splenocytes and tumor cells: anti-CD45 (Biolegend, Cat# 103116), CD3 (Biolegend, Cat# 100218), NK1.1 (Biolegend, Cat# 108708), CD49b (Biolegend, Cat# 108910), CD4 (Biolegend, Cat#100422), CD25 (Biolegend, Cat#101904), FOXP3 (Invitrogen, Waltham, MA, Cat#17-5773-82), CD44 (Biolegend, Cat# 103008), CD62L (Biolegend, Cat#104412), and CD8a (Biolegend, Cat#100706). A CANTO II flow cytometer (BD Bioscience, Franklin Lakes, NJ, USA) was used for cell acquisition, and FlowJo software version 10.8.1 (TreeStar, San Carlos, CA, USA) was used for analysis. The detailed information about the dilutions and catalog numbers of the antibodies is provided in Table 1.

### 2.11. Quantitative PCR (qPCR) Analysis

The total RNA extraction was conducted from cancer cells using an RNeasy Plus Mini Kit (Qiagen, Germantown, MD, USA), and reverse transcription was conducted using a PrimeScript First Strand cDNA Synthesis Kit (Takara, Mountain View, CA, USA). The qPCR analysis was performed using SYBR Premix Ex Taq (Tli RNase H Plus) (Takara) and a CFX384 Touch instrument (Bio-Rad, Hercules, CA, USA). The primers used here were as follows: Ang1, FW 5′-CATTCTTCGCTGCCATTCTG-3′ and Rev 5′-GCACATTGCCCATGTTGAATC-3′; Ang2, FW 5′-TTAGCACAAAGGATTCGGACAAT-3′ and Rev 5′-TTTTGTGGGTAGTACTGTCCATTCA-3′. The PCR conditions used were as follows: initial denaturation was performed at 95 °C for 1 min; 35 cycles of denaturation (95 °C for 30 s), annealing (60 °C for 30 s), and extension (72 °C for 1 min) were performed; and a final extension at 72 °C for 1 min was performed. The detailed information about the primers and thermal conditions is provided in Table 2.

### 2.12. Statistical Analysis

The statistical significance among treated groups was determined by Tukey’s multiple comparisons test using one-way or two-way ANOVA or unpaired *t*-tests (Prism v9, GraphPad, San Diego, CA, USA). The statistical details are provided in the figure legends.

## 3. Results

### 3.1. Lactococcus lactis GEN3013 Induces Cell Death in Various Cancer Cell Types

To identify *L. lactis* strains with cytotoxicity against cancer, a cell viability assay using an MC38 cancer cell line was conducted with six different *L. lactis* strains, including strains that are commercially available and strains that were isolated from human stools in-house. Interestingly, each bacterial lysate, despite being classified into the same species, exerted differential cytotoxicity (Appendix A). The most effective strain was *L. lactis* GEN3013, which was obtained from a healthy donor. To confirm whether *L. lactis* GEN3013 could affect the growth of various cancer cell types, several human cancer cell line (Figure 1A) cultures were supplemented with *L. lactis* GEN3013 lysates. Compared with that of the PBS control (a negative control), *L. lactis* GEN3013 decreased the cell viability dose-dependently (Figure 1B and Appendix A). When *L. lactis* GEN3013 lysates were added to mouse cancer cell line cultures, it reduced the cancer cell viability dose-dependently (Figure 1C). Thus, *L. lactis* GEN3013 induced the cell death of various human and mouse cancer cells by interacting with them. We also confirmed that the conditioned bacterial supernatants of *L. lactis* GEN3013 could reduce the viability of MC38 cancer cells in a dose-dependent manner (Appendix A), indicating that *L. lactis* GEN3013 releases anticancer metabolites.

To elucidate potential anticancer metabolites derived from *L. lactis* GEN3013, we conducted metabolic profiling of the *L. lactis* GEN3013-derived metabolites. We discovered anticancer metabolites derived from the *L. lactis* GEN3013 cell lysates (Table 3). Some metabolites, such as vaccenic acid, s-adenosylmethionine, and oleic acid, could inhibit cancer cell growth and viability [35,36,37], and other metabolites, including methylthioadenosine, citrate, and betaine, could induce cancer cell apoptosis [38,39,40]. N, N-dimethylformamide inhibits the tumor invasion of tumor-associated macrophages by reducing ROS production [41]. GABA increases oxaliplatin sensitivity in human colon cancer cells [42] (Appendix A). Based on previous studies, *L. lactis* GEN3013 may induce cancer cell death through its antitumor metabolites.

### 3.2. L. lactis GEN3013 Administration Inhibits Tumor Growth by Directly Affecting Cancer Cells

To understand whether *L. lactis* GEN3013 directly inhibited tumor growth in vivo similarly to the results found with in vitro assays, CT26 (mouse colon cancer) and 4T1 (mouse breast cancer) cells were subcutaneously inoculated into immune-deficient *Balb/c* nude mice, and then the mice were orally administered *L. lactis* GEN3013 daily. This showed that *L. lactis* GEN3013 significantly suppressed the tumor growth in both cell lines (Figure 2A). Similarly, *L. lactis* GEN3013 suppressed HCT116 human colon cancer cell growth in an immunodeficient mouse model (Figure 2B). Therefore, *L. lactis* GEN3013 could be used as a novel anticancer agent based on its direct killing effect on cancer cells.

To determine how *L. lactis* GEN3013 inhibits tumor growth in detail, we examined the cancer cell characteristics that were influenced by *L. lactis* GEN3013 treatment. Cancer cell migration is a typical process that is associated with tumor invasion and metastasis from the primary tumor to distant organs. For this reason, it was tested whether *L. lactis* GEN3013 affects the mobility of HCT116 and A549 cancer cells using a wound-healing assay and ImageJ processing analysis [43]. The mobility of HCT116 cancer cells markedly declined after *L. lactis* GEN3013 lysate treatment (13.50% ± 8.55%) compared to that after PBS treatment (33.82% ± 5.86%) (Figure 2C). In comparison to PBS treatment (65.31% ± 1.69%), the mobility of A549 cancer cells was markedly reduced after *L. lactis* GEN3013 lysate treatment (17.60% ± 5.19%) (Figure 2D). Treating MC38 cancer cells with the conditioned bacterial supernatant of *L. lactis* GEN3013 (26.91% ± 6.44%) also decreased the mobility of the cancer cells compared to the DMEM-MRS mixture control (43.59% ± 9.96%) and the no-treatment control (46.98% ± 7.62%) (Appendix A).

Another crucial phenotype of cancer progression is angiogenesis triggered by tumor-related signals, such as vascular endothelial growth factor (VEGF) and angiopoietin-1 and -2 (Ang1, Ang2). VEGF is involved in vasculogenesis and angiogenesis signaling. Furthermore, Ang1 is a critical factor contributing to blood vessel maturation. In contrast, Ang2 mediates tumor metastasis by enhancing vessel permeability to the point of blood vessel disruption, leading to leaky vessels. The leaky vessels enable cancer cells to penetrate the blood vessels, which is a critical step for the migration to distal organs. Therefore, we examined the expression of angiogenesis-related genes in *L. lactis* GEN3013-treated HCT116 cancer cells. *L. lactis* GEN3013 reduced the expression of VEGF, Ang1, and Ang2 (Figure 2E,F). These results suggest that *L. lactis* GEN3013 can prevent tumor growth by inhibiting cancer cell migration and angiogenesis.

### 3.3. L. lactis GEN3013 Enhances the Chemotherapy Therapeutic Effects

Oxaliplatin and pemetrexed have been approved by the US Food and Drug Administration as colorectal and lung cancer medication, respectively. Despite their effectiveness in treating both naïve and relapsed advanced cancer, a narrow therapeutic index limits their use; therefore, the development of combinatorial agents that can either enhance their efficacy or reduce the dose-limiting toxicity is needed. We tested whether the therapeutic effects of oxaliplatin or pemetrexed were enhanced by combination treatment with *L. lactis* GEN3013. The proliferation of human cancer cell lines (HCT116, colon; LoVo, colon; A549, lung) and murine cancer cell lines (MC38, colon; LLC1, lung) was inhibited by a single treatment with *L. lactis* GEN3013 and oxaliplatin or pemetrexed (Figure 3A,B). Moreover, a remarkable inhibitory effect was observed in the cancer cells treated with *L. lactis* GEN3013 and the anticancer drugs compared to the effect in those treated only with *L. lactis* GEN3013 or the anticancer drugs (Figure 3A,B). In the case of oxaliplatin, the dose limitation was overcome by co-treatment with *L. lactis* GEN3013 because it enhanced the cancer-killing effect at all concentrations in HCT116 and LoVo human colon cancer cells (Figure 3C). The combined effect of *L. lactis* GEN3013 and oxaliplatin was validated in a syngeneic mouse model. After prevention treatment with *L. lactis* GEN3013 for two weeks, the mice were subcutaneously inoculated with MC38 cancer cells. Then, the mice were orally administered *L. lactis* GEN3013 daily, and oxaliplatin was intra-peritoneally injected twice a week. The results showed that the anticancer effect of the *L. lactis* GEN3013 treatment alone was similar to that of the oxaliplatin treatment alone, and a synergetic effect occurred for the treatment with both *L. lactis* GEN3013 and oxaliplatin (Figure 3D).

### 3.4. Combination of L. lactis GEN3013 and Chemotherapy Reprograms the Host Immune System

Oxaliplatin induces immunogenic death in cancer cells [44,45], which is caused by chronic exposure to damage-associated molecular patterns. Immunogenic cell death can contribute to host antitumor immunity [46,47]. To examine whether antitumor immunity is enhanced by *L. lactis* GEN3013 treatment, we observed immune cell population changes in the spleen and tumor tissues. The tissues were dissected from MC38 tumor-bearing mice treated with control IgG, *L. lactis* GEN3013, oxaliplatin, and *L. lactis* GEN3013 with oxaliplatin. The levels of essential immune cells for antitumor immunity, such as CD4^+^ T cells, CD8^+^ effector T cells, and NK cells, were increased by *L. lactis* GEN3013 administration compared to those obtained via control IgG treatment in both the spleen and tumor microenvironments. In contrast, the number of regulatory T cells, known as suppressor T cells, was significantly decreased by the *L. lactis* GEN3013 treatment. These immune-boosting responses were significantly increased in the group that was administered *L. lactis* GEN3013 with oxaliplatin (Figure 4A,B). Therefore, *L. lactis* GEN3013 may be a suitable anticancer agent for combination therapy with chemotherapy to improve its therapeutic efficacy. 

### 3.5. L. lactis GEN3013 Enhances the Therapeutic Effects of Immunotherapy

The discovery of immune checkpoints, such as PD-1, CTLA-4, LAG-3, and TIM-3, has shifted the cancer treatment paradigm by targeting cancer itself to modulate a tumor-associated immune response. Despite the dramatic antitumor efficacy of immunotherapy, almost 70–80% of patients do not respond to treatment. Specific bacterial strains improve the responsiveness to immune checkpoint inhibitors, which led us to analyze the antitumor immunity of *L. lactis* GEN3013. After isolating mouse CD4^+^ and CD8^+^ T cells from the spleen, the T cells were stained with CFSE dye to monitor cell proliferation. The conditioned bacterial supernatant of *L. lactis* GEN3013 was then added to each well and incubated with T cells for 3 days. Peak 0 (blue population) represents no proliferative cells, whereas peaks 3, 4, and 5 (red population) indicate actively proliferative cells (Figure 5A,B). In both CD4^+^ and CD8^+^ T cells, the static population decreased and the proliferative population increased after *L. lactis* GEN3013 treatment (Figure 5A,B). To confirm whether it was a *L. lactis* GEN3013-specific effect, two additional species (*Eubacterium biforme* and *Coprococcus catus*), which are known to be harmless to humans, were tested by a T cell proliferation assay. As with *L. lactis* GEN3013, the assay was conducted using RPMI-conditioned bacterial supernatants (Appendix A). *E. biforme* had no significant impact on the T cell proliferation, while *C. catus* suppressed the T cell proliferation, compared to the only-act control (Appendix A). This showed that not all bacteria species can activate or promote T cell proliferation; this is *L. lactis* GEN3013’s distinctive ability.

We tested whether *L. lactis* GEN3013 could boost the activity of human T cells. IFN-γ is an immune-activating cytokine secreted by cytotoxic T cells and is responsible for inhibiting tumor growth. IL-10 is an immune-suppressing cytokine secreted by regulatory T cells that is capable of disturbing cytotoxic T cell activity. *L. lactis* GEN3013 treatment increased the IFN-γ level by more than two times and decreased the IL-10 production by T cells compared to that of the negative control with *E. coli* (Figure 5C and Appendix A). These results indicate that *L. lactis* GEN3013 can boost immunity against tumors. Therefore, we confirmed that *L. lactis* GEN3013 can improve the effectiveness of conventional immunotherapy. In the same syngeneic mouse model system, a combination treatment with *L. lactis* GEN3013 and the PD-1 blockade reduced tumor growth more than the *L. lactis* GEN3013 treatment alone (Figure 5D,E). This indicates that *L. lactis* GEN3013 could help overcome limited responses to immunotherapy.

### 3.6. The Combination of L. lactis GEN3013 and Immunotherapy Reprograms the Host Immune System

The responses to immune checkpoint (ICP) inhibitors are influenced by various factors, including the tumor’s mutational burden, neoantigens, PD-L1 expression, and tumor-infiltrating immune cells [48,49,50]. The presence of tumor-infiltrating lymphocytes (TILs) is a key factor for ICP inhibitor effectiveness [51,52]. Studies have found a longer progression-free survival or overall survival when high CD8^+^ TIL levels are present in patients with several cancer types [53,54]. In other studies, a specific subset of CD4^+^ T cells was associated with patient survival, and any loss of cells from this subset was associated with resistance to ICP inhibitor therapy [55,56]. Therefore, ICP inhibitors are believed to be more effective when cytotoxic immune cells infiltrate and are activated in the tumor microenvironment. We assessed the immune-boosting effects of *L. lactis* GEN3013 when administered in combination with immunotherapy. The co-treatment with *L. lactis* GEN3013 and the PD-1 blockade significantly increased the number of systemic immune cells in the spleen and the number of tumor-infiltrating T cells and NK cells in the tumor microenvironment (Figure 6A,B). These results indicate that *L. lactis* GEN3013 reprograms the host immune system to improve antitumor immunity when used alone or in combination with conventional immunotherapy. Overall, *L. lactis* GEN3013 may be a suitable anticancer agent for combination therapy with immunotherapy to improve its therapeutic effects.

## 4. Discussion

Cancer remains a major cause of death worldwide, accounting for nearly 10 million deaths in 2020 [57]. Therefore, appropriate and effective cancer treatments are required to improve the probability of survival and therapeutic efficacy. The gut microbiota is a key factor in the response to cancer therapy, since many studies have implied that gut dysbiosis contributes to carcinogenesis [58,59] and drug responsiveness [17,60]. Additionally, specific bacterial species and strains, but not all microbiota, have distinct antitumor immunity by modulating the systemic function or the function of TILs [61,62,63]. *L. lactis* GEN3013 showed antitumor effects by directly inhibiting the proliferation of cancer cells (Figure 1) and by improving host immunity (Figure 4 and Figure 6). Thus, *L. lactis* GEN3013 might be a promising antitumor agent and useful for combination therapy with other conventional cancer therapies.

Oxaliplatin and pemetrexed are well-known chemotherapy agents used for colon cancer and non-small cell lung cancer, respectively. These drugs have a narrow therapeutic index, implying that small changes in the drug dose could lead to serious side effects and treatment failure [64,65,66]. Oxaliplatin leads to dose-dependent toxicity in the hematopoietic and nervous systems, inducing anemia, thrombocytopenia, peripheral neuropathy, and pain [67]. Pemetrexed also has a dose limitation for cancer treatment because it can cause gastrointestinal side effects, such as nausea, vomiting, stomatitis, and diarrhea [68]. Up to 87% of patients given chemotherapy experienced adverse events during or after treatment [69,70]. Therefore, other modalities, such as the microbiota, are emerging to improve the efficacy of chemotherapy. A study revealed that *Akkermansia muciniphila* potentiated the antitumor efficacy of oxaliplatin through a specific branched-chain amino acid (BCAA) metabolite [71]. Here, *L. lactis* GEN3013 was identified as a reasonable combination treatment with chemotherapy because it improved the therapeutic effect of oxaliplatin (Figure 3).

Although immunotherapy has emerged as a powerful cancer treatment, only a minority of patients (20–40%) have responded to PD-1/PD-L1 blockade therapy [51,72,73]. Cytotoxic anticancer agents are frequently combined with immunotherapy to improve responsiveness, since these agents can be used to treat tumors with more mutations [74,75,76]. Recently, the microbiome has been considered a good immunotherapy combination agent because the gut microbiota can contribute to the host immune system, both locally and systemically. *Bifidobacterium* facilitates anti-PD-L1 efficacy by promoting antitumor immunity [17], and *Bacteroides fragilis* has been associated with CTLA-4 blockade efficacy [77]. Hence, we expected that *L. lactis* GEN3013 would enhance the antitumor efficacy of immunotherapy. It was validated that *L. lactis* GEN3013 boosted the host immune system (Figure 6) and increased the efficacy of the PD-1 blockade in a syngeneic mouse model (Figure 5D). Therefore, *L. lactis* GEN3013 is a promising anticancer agent for combination treatment with conventional immunotherapy.

Bevacizumab is a well-known anticancer agent used to treat several types of cancer, including colon, lung, and renal cancer. It is a monoclonal antibody that inhibits angiogenesis by suppressing VEGF-A expression. Despite its therapeutic effects, bevacizumab has severe side effects, such as gastrointestinal perforation, bleeding, and allergic reactions, and a limited response rate of approximately 40–50%. *L. lactis* GEN3013 may be used as a supplement to anti-angiogenesis agents, such as bevacizumab, because it directly reduces the expression of angiogenesis factors and the migration of cancer cells (Figure 2C–F).

Overall, we demonstrated that *L. lactis* GEN3013 could be used as a novel anticancer agent in monotherapy or combination therapy with chemotherapy or immunotherapy. However, there are limitations to this study. Despite discovering candidate metabolites of *L. lactis* GEN3013, (i) we did not identify the metabolites mainly responsible for the antitumor immunity. To understand the exact anticancer mechanism of *L. lactis* GEN3013, it is necessary to assess the effects of metabolites derived from *L. lactis* GEN3013 on cancer growth and immune activation through further investigation. These further experiments could reveal the elaborate molecular mechanisms of *L. lactis* GEN3013. Although we confirmed the *L. lactis* GEN3013 combination effects with oxaliplatin or PD-1 blockade, (ii) we did not observe the combination effects with pemetrexed or bevacizumab. In further research, we will demonstrate whether *L. lactis* GEN3013 can overcome the limitations of additional anticancer drugs by evaluating its combination effects.

## 5. Conclusions

In summary, we validated that *L. lactis* GEN3013 can be used as a live biotherapeutic agent without any genetic alterations. Because *L. lactis* GEN3013 reduced the proliferation of various cancer cells, we propose that *L. lactis* GEN3013 could be used as a monotherapy for cancer. Moreover, *L. lactis* GEN3013 could be utilized as a combination therapy agent with immunotherapy or chemotherapy. This is because it was confirmed that *L. lactis* GEN3013 improved the efficacy of conventional anticancer therapies in a syngeneic mouse model. Although we confirmed that *L. lactis* GEN3013 showed anticancer immunity, further studies on the elaborate molecular mechanisms of *L. lactis* GEN3013 are required for its development as an anticancer drug.

## Figures and Tables

**Figure 1 cancers-14-04083-f001:**
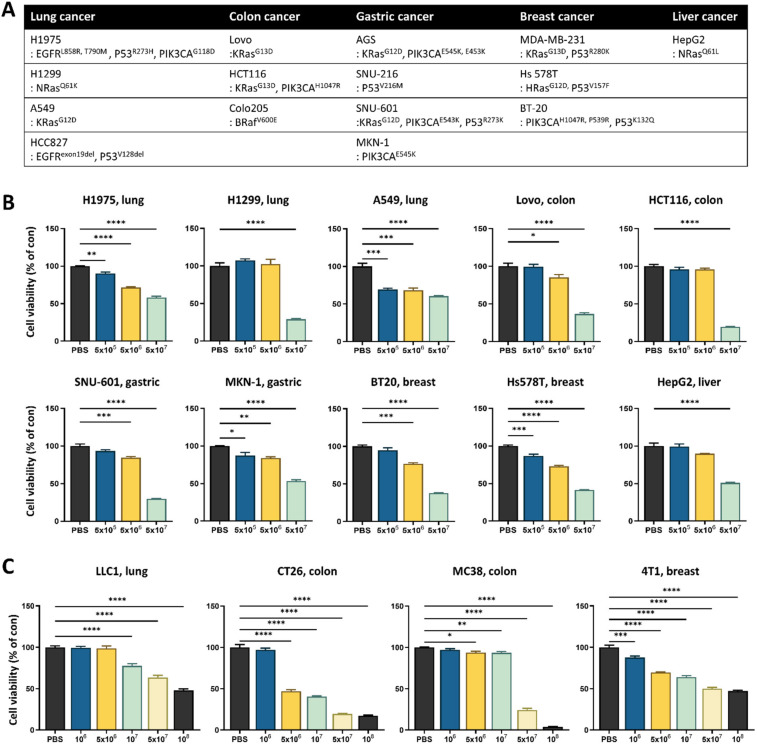
*Lactococcus lactis* GEN3013 induces the death of various cancer cells. (**A**) Human cancer cell lines were organized according to the cancer type and genetic characteristics. (**B**,**C**) Various human and mouse cancer cell lines were treated with 0.5% *L. lactis* GEN3013 dose-dependently and incubated for 72 h. Cell viability was measured using an MTT assay and calculated relative to a PBS control. All human and mouse cancer cells showed significant *p*-values (*p* ≤ 0.0001) according to one-way ANOVA tests. Data represent relative cell viability ± SEM; the *p*-values were calculated using one-way ANOVA with Tukey’s multiple comparison test compared to a PBS control; * *p* ≤ 0.05, ** *p* ≤ 0.01, *** *p* ≤ 0.001, and **** *p* ≤ 0.0001.

**Figure 2 cancers-14-04083-f002:**
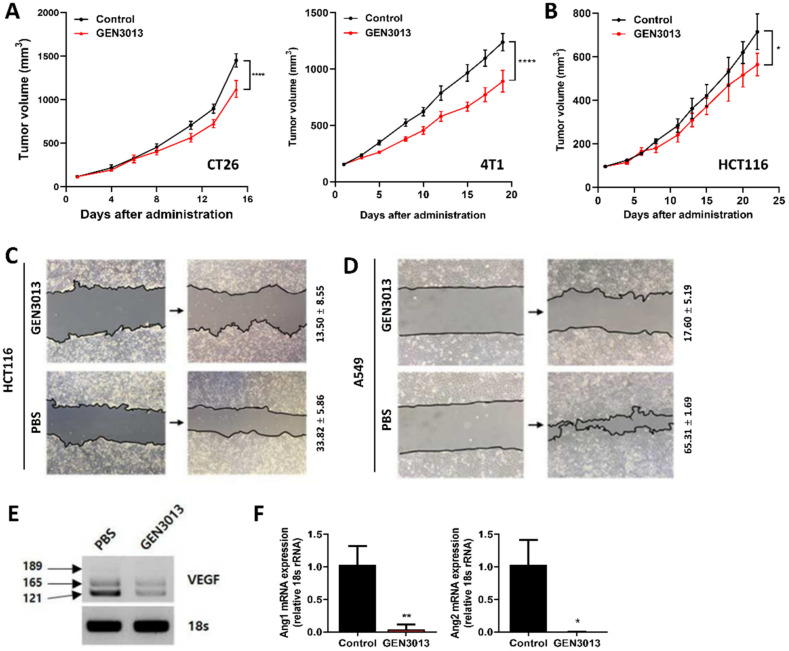
*Lactococcus lactis* GEN3013 treatment prevents tumor growth through direct effects on cancer cells. (**A**) Nude mice were inoculated with mouse cancer cell lines, CT26 (N = 7 per group) and 4T1 (N = 10 per group), respectively, 14 days after the initial oral administration of *L. lactis* GEN3013. While *L. lactis* GEN3013 was orally administered daily to mice, the tumor growth was measured three times a week. *L. lactis* GEN3013 versus control: CT26, *p* ≤ 0.0001; 4T1, *p* ≤ 0.0001. (**B**) Nude mice were inoculated with HCT116 human cancer cells (N = 5 per group). *L. lactis* GEN3013 versus control: *p* = 0.0239. (**C**,**D**) The mobility of HCT116 cells or A549 cells was measured using a wound-healing assay and calculated relative to the status of the zero time point of cell damage. (**C**) *L. lactis* GEN3013 treatment, 13.50% ± 8.55%; PBS treatment, 33.82% ± 5.86%. (**D**) *L. lactis* GEN3013 treatment, 17.60% ± 5.19%; PBS treatment, 65.31% ± 1.69%. (**E**) VEGF expression in HCT116 cells was analyzed using a western blot. The levels of VEGF isoforms, including the 121, 165, and 189 isoforms, were detected. (**F**) The Ang1 and Ang2 mRNA expression was measured using quantitative PCR. *L. lactis* GEN3013-treated versus untreated: Ang1, *p* = 0.0047; Ang2, *p* = 0.0136. For all graphs, data are shown as the mean ± SEM and *p*-values are represented as * *p* ≤ 0.05, ** *p* ≤ 0.01, and **** *p* ≤ 0.0001. Statistical significance was determined using two-way ANOVA with Tukey’s test for multiple comparisons (**A**,**B**), or a two-sided unpaired *t*-test (**F**).

**Figure 3 cancers-14-04083-f003:**
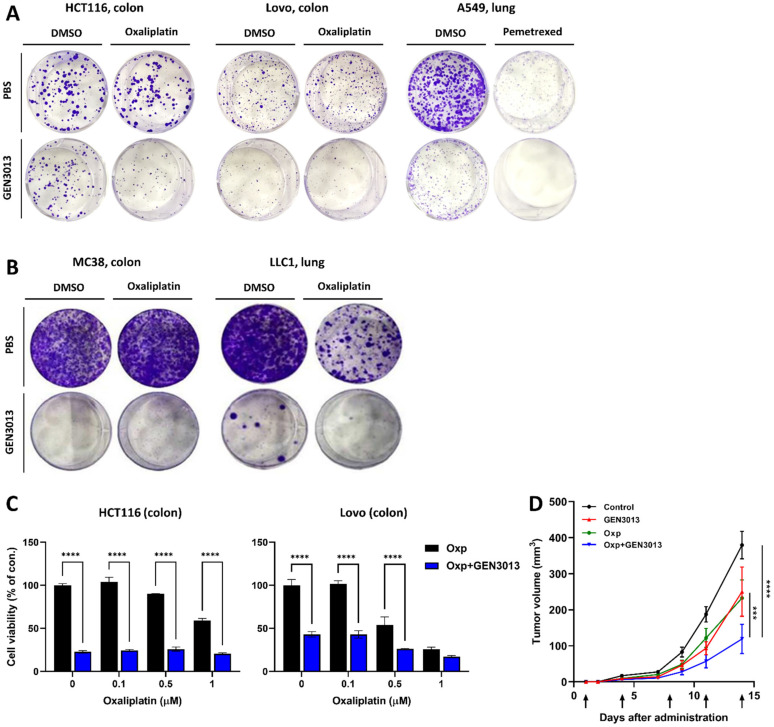
*Lactococcus lactis* GEN3013 enhances the therapeutic effects of chemotherapy. (**A**,**B**) Cancer cell growth inhibition was observed using crystal violet staining. HCT116 and LoVo human colon cancer cells were treated with 0.1 μM oxaliplatin and A549 human lung cancer cells were treated with 0.05 μM pemetrexed. MC38 and LLC1 mouse cancer cell lines were cultured with 2 μM oxaliplatin. (**C**) The cell viability was measured after treating HCT116 and LoVo cancer cells with both oxaliplatin and *L. lactis* GEN3013. Oxaliplatin was administered dose-dependently (0, 0.1, 0.5 and 1 μM), while the *L. lactis* GEN3013 dose was fixed at 0.5%. All concentrations of oxaliplatin showed increased efficacy in combination with *L. lactis* GEN3013 in both HCT116 and LoVo cells (oxaliplatin versus oxaliplatin + *L. lactis* GEN3013, *p* ≤ 0.0001 at all concentrations, except for 1 μM oxaliplatin in LoVo cells). (**D**) C57BL/6 mice were inoculated with MC38 mouse cancer cells (N = 6–8 per group) 14 days after the initial oral administration of *L. lactis* GEN3013. While *L. lactis* GEN3013 was orally administered daily, oxaliplatin was intraperitoneally injected at 1, 4, 8, 11 and 14 days after tumor inoculation. Control versus oxaliplatin, *p* ≤ 0.0001; control versus *L. lactis* GEN3013, *p* ≤ 0.0001; control versus oxaliplatin + *L. lactis* GEN3013, *p* ≤ 0.0001; oxaliplatin versus oxaliplatin + *L. lactis* GEN3013, *p* = 0.0002; *L. lactis* GEN3013 versus oxaliplatin + *L. lactis* GEN3013, *p* = 0.0001, oxaliplatin versus *L. lactis* GEN3013, ns. For all graphs, data are shown as the mean ± SEM and *p*-values are represented as *** *p* ≤ 0.001 and **** *p* ≤ 0.0001. The statistical significance was determined using two-way ANOVA and Tukey’s test for multiple comparisons.

**Figure 4 cancers-14-04083-f004:**
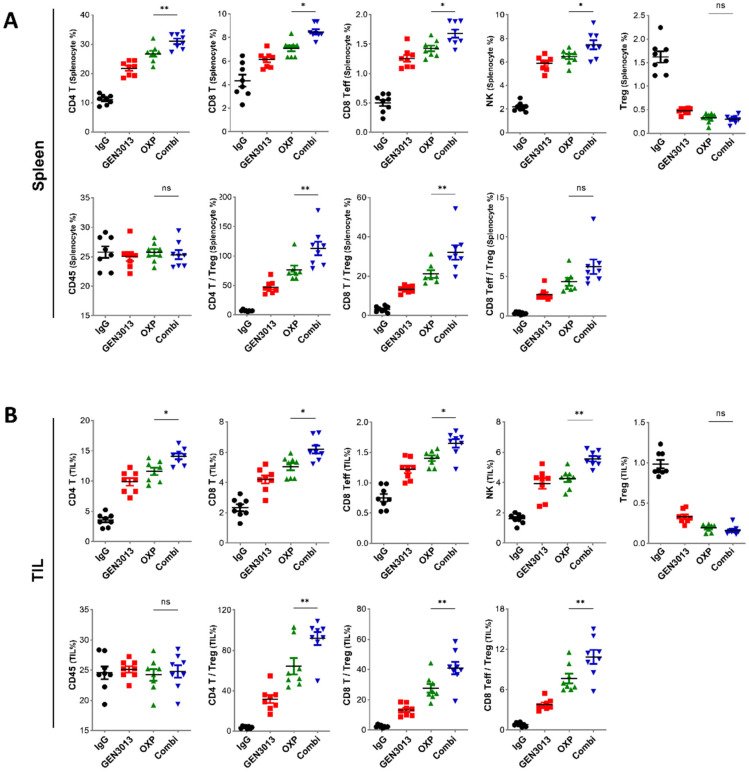
Combination of *Lactococcus lactis* GEN3013 and chemotherapy reprograms the host immune system. (**A**,**B**) Changes in the immune cell population of the spleen (**A**) or the tumor microenvironment (**B**) were measured via immune cell profiling after the administration of *L. lactis* GEN3013, oxaliplatin, or their combination (N = 8 per each group). Oxaliplatin versus combination in the spleen tissues: CD4^+^ T cells, *p* = 0.0075; CD8^+^ T cells, *p* = 0.0238; effector CD8^+^ T cells, *p* = 0.0213; NK cells, *p* = 0.0455; Tregs, ns; CD45^+^ cells, ns; CD4^+^ T cells/Tregs ratio, *p* = 0.007; CD8^+^ T cells/Tregs ratio, *p* = 0.0069; effector CD8^+^ T cells/Tregs ratio, ns. Oxaliplatin versus combination in the tumor tissues: CD4^+^ T cells, *p* = 0.012; CD8^+^ T cells, *p* = 0.0155; effector CD8^+^ T cells, *p* = 0.0386; NK cells, *p* = 0.0022; Tregs, ns; CD45^+^ cells, ns; CD4^+^ T cells/Tregs ratio, *p* = 0.0087; CD8^+^ T cells/Tregs ratio, *p* = 0.0058; effector CD8^+^ T cells/Tregs ratio, *p* = 0.0085. For all graphs, data are shown as the mean ± SEM and *p*-values are represented as * *p* ≤ 0.05, ** *p* ≤ 0.01, and ns (not significant). The statistical significance was determined using one-way ANOVA and Tukey’s test for multiple comparisons.

**Figure 5 cancers-14-04083-f005:**
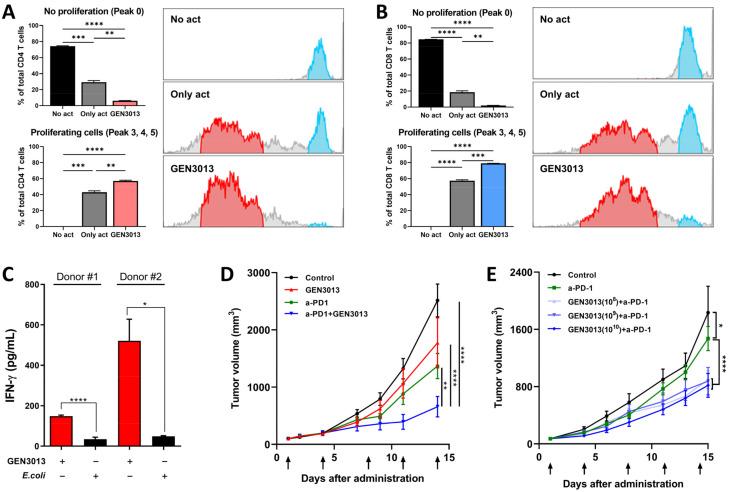
*Lactococcus lactis* GEN3013 enhances the therapeutic effects of immunotherapy. (**A**,**B**) Mouse T cell proliferation was measured using CFSE staining and flow cytometry. Blue population (right panel) represents no proliferative cells (peak 0), while the red population indicates actively proliferative cells (peaks 3, 4, and 5). The *p*-values for the peak 0 population in CD4^+^ T cells (*p* = 0.0002, *p* ≤ 0.0001, *p* = 0.0019), the peak 3, 4, and 5 populations in CD4^+^ T cells (*p* = 0.0003, *p* ≤ 0.0001, *p* = 0.0073), the peak 0 population in CD8^+^ T cells (*p* ≤ 0.0001, *p* ≤ 0.0001, *p* = 0.0033), and the peak 3, 4, and 5 populations in CD8^+^ T cells (*p* ≤ 0.0001, *p* ≤ 0.0001, *p* = 0.0004) were in the following order: no act versus only act, no act versus *L. lactis* GEN3013, and only act versus *L. lactis* GEN3013. (**C**) Human T cell activation was observed via ELISA of IFN-γ. *E. coli* versus *L. lactis* GEN3013: donor #1, *p* = 0.0001; donor #2, *p* = 0.0105. (**D**) C57BL/6 mice were inoculated with MC38 mouse cancer cells (N = 6 per group) 14 days after the initial oral administration of *L. lactis* GEN3013. *L. lactis* GEN3013 was orally administered daily, whereas the PD-1 blockade was intraperitoneally injected at 1, 4, 8, 11, and 14 days after tumor inoculation. Control versus PD-1 blockade, *p* ≤ 0.0001; control versus *L. lactis* GEN3013, *p* = 0.0019; control versus PD-1 blockade + *L. lactis* GEN3013, *p* ≤ 0.0001; PD-1 blockade versus PD-1 blockade + *L. lactis* GEN3013, *p* = 0.0051; *L. lactis* GEN3013 versus PD-1 blockade + *L. lactis* GEN3013, *p* ≤ 0.0001; PD-1 blockade versus *L. lactis* GEN3013, ns. (**E**) *L. lactis* GEN3013 was orally administered dose-dependently at 10^8^, 10^9^, and 10^10^ CFUs (N = 7 per group). Control versus PD-1 blockade, *p* = 0.0318; control versus *L. lactis* GEN3013 (all concentrations), *p* ≤ 0.0001; PD-1 blockade versus PD-1 blockade + *L. lactis* GEN3013 (all concentrations), *p* ≤ 0.0001. For all graphs, data are shown as the mean ± SEM and *p*-values are represented as * *p* ≤ 0.05, ** *p* ≤ 0.01, *** *p* ≤ 0.001, and **** *p* ≤ 0.0001. Statistical significance was determined using one-way (**A**,**B**) or two-way (**D**,**E**) ANOVA with Tukey’s test for multiple comparisons, or an unpaired *t*-test (**C**).

**Figure 6 cancers-14-04083-f006:**
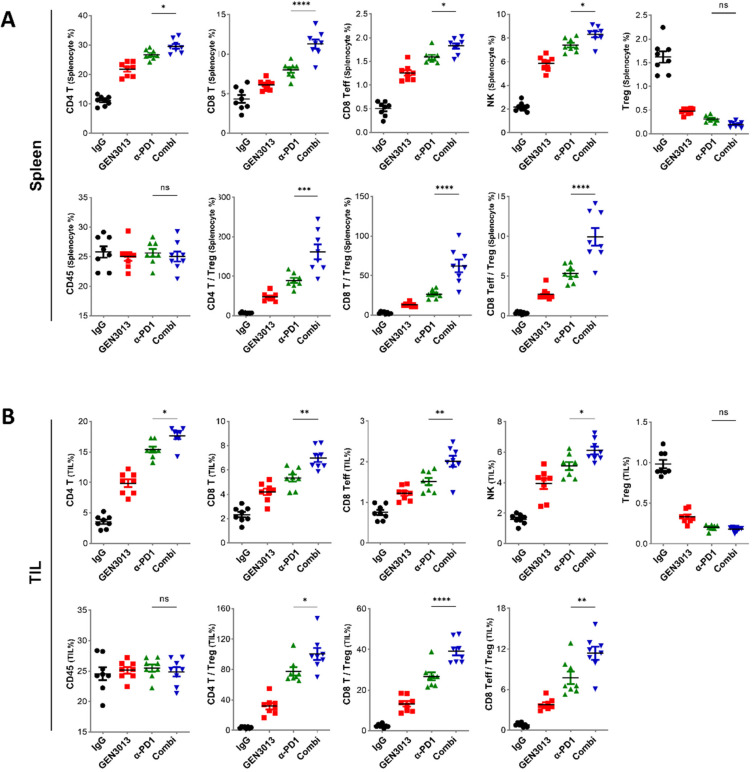
Combination of *Lactococcus lactis* GEN3013 and immunotherapy reprograms the host immune system. (**A**,**B**) Changes in the immune cell population of the spleen (**A**) or the tumor microenvironment (**B**) were measured via immune cell profiling after the administration of *L. lactis* GEN3013, the PD-1 blockade, or their combination (N = 8 per each group). PD-1 blockade versus combination in the spleen tissues: CD4^+^ T cells, *p* = 0.0419; CD8^+^ T cells, *p* ≤ 0.0001; effector CD8^+^ T cells, *p* = 0.0257; NK cells, *p* = 0.0182; Tregs, ns; CD45^+^ cells, ns; CD4^+^ T cells/Tregs ratio, *p* ≤ 0.0001; CD8^+^ T cells/Tregs ratio, *p* ≤ 0.0001; effector CD8^+^ T cells/Tregs ratio, *p* ≤ 0.0001. PD-1 blockade versus combination in tumor tissues: CD4^+^ T cells, *p* = 0.019; CD8^+^ T cells, *p* = 0.0014; effector CD8^+^ T cells, *p* = 0.0031; NK cells, *p* = 0.0408; Tregs, ns; CD45^+^ cells, ns; CD4^+^ T cells/Tregs ratio, *p* = 0.021; CD8^+^ T cells/Tregs ratio, *p* ≤ 0.0001; effector CD8^+^ T cells/Tregs ratio, *p* = 0.0045. For all graphs, data are shown as the mean ± SEM and *p*-values are represented as * *p* ≤ 0.05, ** *p* ≤ 0.01, *** *p* ≤ 0.001, **** *p* ≤ 0.0001, and ns (not significant). Statistical significance was determined using one-way ANOVA with Tukey’s test for multiple comparisons.

**Table 1 cancers-14-04083-t001:** The dilutions and catalog numbers of the antibodies of flow cytometry.

Target	Conjugated Fluorescent	Company	Catalog Number	Usage Dilution
CD45	APC/Cyanin7	Biolegend	103116	1/40
CD3	PerCP/Cyanine5.5	Biolegend	100218	1/20
NK1.1	PE	Biolegend	108708	1/40
CD49b	APC	Biolegend	108910	1/40
CD4	PE/Cyanine7	Biolegend	100422	1/100
CD25	PE	Biolegend	101904	1/100
FOXP3	APC	Invitrogen	17-5773-82	1/40
CD44	PE	Biolegend	103008	1/40
CD62L	APC	Biolegend	104412	1/40
CD8a	FITC	Biolegend	100706	1/40

**Table 2 cancers-14-04083-t002:** The primers and thermal conditions of qPCR analysis.

Target	Sequence		PCR Thermal Conditions		
Ang1	FW	5′-CATTCTTCGCTGCCATTCTG-3′	95 °C	1 min	Initial denaturation
	Rev	5′-GCACATTGCCCATGTTGAATC-3′	95 °C	30 s	35 cycles
			60 °C	30 s	
Ang2	FW	5′-TTAGCACAAAGGATTCGGACAAT-3′	72 °C	1 min	
	Rev	5′-TTTTGTGGGTAGTACTGTCCATTCA-3′	72 °C	1 min	Final extension

**Table 3 cancers-14-04083-t003:** *Lactococcus lactis* GEN3013-derived metabolites.

	Composition of MRS Media	Bacterial Lysate	*p*-Value
Vaccenic acid	0 ± 0	6.834 ± 0.023	<0.001
S-Adenosylmethionine	0 ± 0	6.759 ± 0.050	<0.001
Oleic acid	0 ± 0	5.014 ± 0.041	<0.001
N,N-Dimethylformamide	0 ± 0	5.206 ± 0.057	<0.001
Methylthioadenosine	0 ± 0	7.305 ± 0.027	<0.001
Indole lactic acid	0 ± 0	7.442 ± 0.088	<0.001
GABA (gamma-aminobutyric acid)	0 ± 0	5.425 ± 0.068	<0.001
Citrate	0 ± 0	6.443 ± 0.085	<0.001
Betaine	0 ± 0	5.802 ± 0.102	<0.001

Normalized log10 and 5 technical replicates; statistical analysis was determined by Student’s *t*-test. Data are expressed as means ± SEMs.

## Data Availability

The data that support the findings of this study are available from the corresponding author, H.P., upon reasonable request.

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
