# Peer review of "Live Biotherapeutic Lactococcus lactis GEN3013 Enhances Antitumor Efficacy of Cancer Treatment via Modulation of Cancer Progression and Immune System"

_cancers, 2022, doi:10.3390/cancers14174083_

Round 1

Reviewer 1 Report

Kim et al. present a well-characterized study essentially examining the effect of L. lactis homogenate on cancer cell viability and its suitability as a “biotherapeutic agent”. The concept of this research is very interesting and opens to newer and more natural avenues towards discovering therapeutic modalities.

-          My major critique is that while the findings are neat, it begs an important question- is the effect of L. lactis on cancer cell viability biologically meaningful? Bacterial contamination of tissue cultures is a common problem encountered by many labs. In all such cases bacterial contamination is unfavorable for cell growth. While the authors seem to partly explain by testing for other L. lactis strains and demonstrate that not all strains have the capability to restrict cancer cell growth, I would like to see if the authors can utilize normal/non-tumorigenic cell lines such as MCF10A, MCF12A (breast) as well as similar healthy controls in colon and lung and show that the GEN3013 strain does not affect the growth of healthy cells and exerts the specific growth-inhibitory effect only on cancer cells. Unless this can be demonstrated, I feel that one cannot reasonably validate GEN3013 as having a biotherapeutic tumor-suppressor effect. (Line 581/582- The authors go as far as suggesting GEN3013 suitable for monotherapy).  

-          Line 146-150 in materials section are unclear. What is the purpose of extracting bacterial supernatant media here? Are the authors using “bacterial lysates” or bacterial supernatant/conditioned media to do experiments with cancer cells? The authors must clearly describe in the methods section why both were used instead of just one. Also please avoid the term “supernatant media” when referring to bacterial culture media. The more appropriate technical term is “conditioned” media.

-          Avoid the term “co-culture” in this context. Co-culture refers to culturing to “live” cell types together. Instead line 282 can be re-phrased as: “several human cancer cell line cultures were supplemented with GEN3013 lyates”

-          Section 2.5 Metabolic profiling: How were the metabolites “enriched” from the crude extracts? How many replicates were done? This seems like a rather crude metabolic profiling experiment.

-          Please provide a supplementary table of all metabolites identified along with compound names, m/z, ion intensities and p-value to account for an estimate of reproducible detection by MS.

-          There seems to be no justification or explanation for how the authors came up with the series of doses of lysates to use in the viability expts. How can they prove that the growth inhibitory effects seen were not simply due to rampant cytotoxicity which may be observed with an excess of any foreign compound/other bacteria used in tissue cultures.  

-          Lines 291 and 292 are contradictory. Did the authors extract metabolites from “conditioned media” or bacterial “lysates” ?

Author Response

1. My major critique is that while the findings are neat, it begs an important question- is the effect of L. lactis on cancer cell viability biologically meaningful? Bacterial contamination of tissue cultures is a common problem encountered by many labs. In all such cases bacterial contamination is unfavorable for cell growth. While the authors seem to partly explain by testing for other L. lactis strains and demonstrate that not all strains have the capability to restrict cancer cell growth, I would like to see if the authors can utilize normal/non-tumorigenic cell lines such as MCF10A, MCF12A (breast) as well as similar healthy controls in colon and lung and show that the GEN3013 strain does not affect the growth of healthy cells and exerts the specific growth-inhibitory effect only on cancer cells. Unless this can be demonstrated, I feel that one cannot reasonably validate GEN3013 as having a biotherapeutic tumor-suppressor effect. (Line 581/582- The authors go as far as suggesting GEN3013 suitable for monotherapy). 

(Answer)

   Thank you so much for your kind comment. I understood and agreed with your concerns. So, following your advice, we validated the direct-killing effects of L. lactis GEN3013 in a normal cell line. Because the normal breast, colon, and lung cells you recommended could not be obtained immediately, we used the C2C12 cell line instead. The C2C12 is a mouse myoblast cell line that differentiates rapidly, forming contractile myotubes and producing characteristic muscle proteins.

   We seeded 2x104 C2C12 cells in a 96-well cell culture plate, and then treated with L. lactis GEN3013 lysates. Similar to the previous experiment, lysates were treated with 0.5% of the total volume and additionally, 5% lysates were added to confirm whether direct-killing effects of L. lactis GEN3013 were induced by bacterial contamination. Based on the results, the PBS control and L. lactis GEN3013 lysate showed no difference in cell viability of C2C12 cells (Revision figure 1). Therefore, it was concluded that L. lactis GEN3013 is non-toxic to C2C12 cells.

   However, L. lactis GEN3013 cannot be concluded to be bio-safe for all normal cells and tissues. To develop L. lactis GEN3013 as a new drug, the same proof will be required in breast, colon, and lung cells, as Reviewer#1 said. As part of a future study, we will verify the biosafety of L. lactis GEN3013 in various normal cell types. Thank you very much for pointing out an important point.

2. Line 146-150 in materials section are unclear. What is the purpose of extracting bacterial supernatant media here? Are the authors using “bacterial lysates” or bacterial supernatant/conditioned media to do experiments with cancer cells? The authors must clearly describe in the methods section why both were used instead of just one. Also please avoid the term “supernatant media” when referring to bacterial culture media. The more appropriate technical term is “conditioned” media.

(Answer) 

   We apologize for the difficulty in understanding due to the unclear explanation.

   We mainly used bacterial lysates in experiments with cancer cells for the following reasons. The anticancer effect of L. lactis GEN3013 was thought to be due to the bacterial-derived materials that L. lactis GEN3013 produces. In fact, there are many previous studies that show bacterial-derived materials affect human health. For example, bile acids and SCFAs (short-chain fatty acids) derived from bacteria can control metabolic diseases such as type 2 diabetes and NASH [1,2]. In particular, the microbial environment could play a key role in the immune system, as their SCFAs interact with B and T cells to regulate the immune response [3]. In addition, cbDNAs (circulating bacteria DNAs) were widely found among cancer patients, and some papers suggested the possibility of cbDNAs as novel biomarker of cancer diagnosis [4,5]. Thus, it was considered that cbDNAs could not only be used as biomarkers for cancer diseases, but also affect cancer growth and microenvironment. Numerous studies have revealed that microbiota may play a very critical role in various diseases and human health, and metabolites and cbDNAs were key factors in regulating those diseases. In this study, we identified L. lactis GEN3013-derived metabolites, which can exert antitumor activity (Table 3). In conclusion, we determined that bacterial lysates were the most appropriate method for confirming the anticancer effects of L. lactis GEN3013 in vitro assays.

   The reason for using conditioned bacterial supernatants rather than the supernatant itself is as follows. As shown in supplementary figure 2C, MRS-bacterial supernatant was toxic to immune cells because mammalian cell culture media have a pH of 7.5-8.5 whereas bacterial culture media has a pH of 6.5-7.0. In order to reduce differential media-induced toxicity, we prepared conditioned bacterial supernatant after allowing the bacteria to adapt to cell culture media (DMEM or RPMI) for one day. As a result, supplementary figure 2D shows that the toxicity was eliminated.

   Consequently, both bacterial lysates and conditioned bacterial supernatants were used in this study. As reviewed by Reviewer#1, the term "supernatant media" was considered to cause confusion, so it was changed to "conditioned bacterial supernatant" in the manuscript.

3. Avoid the term “co-culture” in this context. Co-culture refers to culturing to “live” cell types Instead line 282 can be re-phrased as: “several human cancer cell line cultures were supplemented with GEN3013 lysates”

(Answer)

   It is very kind of you to point out parts that could be misleading. As you suggested, we edited the sentence.

   Line 295-297: To confirm whether L. lactis GEN3013 could affect the growth of various cancer cell types, several human cancer cell lines (Figure 1A) cultures were supplemented with L. lactis GEN3013 lysates.

4. Section 2.5 Metabolic profiling: How were the metabolites “enriched” from the crude extracts? How many replicates were done? This seems like a rather crude metabolic profiling experiment.

(Answer)

   Although it might be a crude extract, but bacterial cells (1010 CFU) collected by centrifugation at 7000 × g for 5 min at 4 °C were extracted with cold methanol and analyzed with 5 technical replicates to confirm reproducibility (It was added in Table 3 of the manuscript). Coefficient of variation between replication was below 15%.

   Cold methanol was selected as the extraction solvent with reference to the following 2 papers.

   References 1. Prasad Maharjan, R., & Ferenci, T. (2003). Global metabolite analysis: the influence of extraction methodology on metabolome profiles of Escherichia coli. Analytical Biochemistry, 313(1), 145–154. doi:10.1016/s0003-2697(02)00536-5

: As a result of extracting the cell pellet of E. Coli with 6 extraction solutions (Perchloric acid, Alkalikne, Methanol/chloroform, Hot ethanol, Hot methanol, Cold methanol), when comparing the amount of detected metabolites & extraction efficiency, cold methanol was the most (In the case of hot methanol, which is the most efficient, it is not suitable because there is a possibility that the stability of metabolites after extraction is reduced).

   References 2. Park, C., Yun, S., Lee, S.Y. et al. Metabolic Profiling of Klebsiella oxytoca: Evaluation of Methods for Extraction of Intracellular Metabolites Using UPLC/Q-TOF-MS. Appl Biochem Biotechnol 167, 425–438 (2012). https://doi.org/10.1007/s12010-012-9685-9

: The cell pellet of Klebsiella oxytoca was extracted with 6 extraction solutions (perchloric acid, KOH, methanol/chloroform, hot ethanol, hot water, cold methanol) and analyzed by UPLC-QTOF-MS. As a result, metabolites extracted with cold methanol represent stable efficiency and high stability.

   Since the purpose of our global metabolic profiling was detection of all intracellular and cell wall surface metabolites derived from L. lactis GEN3013 cell lysates, we tried to extract various class of metabolites as much as possible based on the above papers.

5. Please provide a supplementary table of all metabolites identified along with compound names, m/z, ion intensities and p-value to account for an estimate of reproducible detection by MS.

(Answer)

   As you comment, compound names, molecular weight, m/z, adducts, retention time (min), and ion intensities of all metabolites detected in L. lactis GEN3013 cell lysates were added to the Supplementary Table 1.

   In previous results, a significant difference was calculated after comparing the L. lactis GEN3013-derived metabolites with the composition of MRS media. In other words, it is said that newly detected metabolites after L. lactis GEN3013 cultivation were identified in this study. The metabolites listed in Table 3 and Supplementary Table 1 have been detected in five biological replicates of L. lactis GEN3013 while the metabolites are not part of MRS media so that we can determine p-value as shown in Table 3. But the more important point is to understand whether metabolites derived from L. lactis GEN3013 are unique when compared to metabolites derived from different L. lactis strains without antitumor effects. Although it was not conducted in this study, as mentioned in the discussion, we will investigate the effectiveness and specificity of L. lactis GEN3013-derived metabolites in further study. That is, we will compare with other bacteria-derived metabolites to confirm whether L. lactis GEN3013-derived metabolites have biological differences.

6. There seems to be no justification or explanation for how the authors came up with the series of doses of lysates to use in the viability expts. How can they prove that the growth inhibitory effects seen were not simply due to rampant cytotoxicity which may be observed with an excess of any foreign compound/other bacteria used in tissue cultures?

(Answer)

   When the bacterial lysates were prepared, the optical density at 600nm (OD600) of the lysates was set to 100. In several studies, it was found that L. lactis would enter a stationary phase when OD600 was 2 or greater, although it depends on the culture conditions. If OD600 is 2 or more, L. lactis have reached the saturated state and are fully metabolically active [6-8]. Hence, we considered the lysates, with OD600 of 100, to be substantially concentrated and to contain a significant amount of bacterial material. Therefore, we treated them with 0.5% of total volume for MTT assay.

   Further, we conducted MTT assays using various bacteria species in a preliminary study. As can be seen below revision figure 2, species A and B showed similar cell viability to the control PBS, while species C showed a significant decrease in cell viability. The result indicated that the changes in viability were simply not due to external substances such as bacterial lysates. Based on the preliminary study, we concluded the growth inhibitory effects of L. lactis GEN3013 were its specific effects.

7. Lines 291 and 292 are contradictory. Did the authors extract metabolites from “conditioned media” or bacterial “lysates”?

(Answer)

   Our apologies for making this difficult to understand. During the sampling process for metabolic analysis, there was a process of discarding the pellet and taking the supernatant after homogenization and centrifugation to remove residual live bacteria. Therefore, the term, supernatant, was used in both biological and technical contexts. When the word, supernatant, was used technically, it may cause confusion. The sentence you mentioned has been clarified as follows.

   Line 307: To elucidate potential anticancer metabolites derived from L. lactis GEN3013, we conducted metabolic profiling of the L. lactis GEN3013-derived metabolites.

Reviewer 2 Report

The authors describe and evaluate the potential use of Lactococcus lactis GEN3013 as an adjuvant agent in anticancer therapies and determine the effect of L. lactis on the viability of various cancer cells. The research is well described, the manuscript is interesting and contains new findings. Figures and tables are well described and informative. And the conclusions follow from the results obtained.

However, the authors could point out that the introduction into the clinic must be preceded by further detailed studies of L. lactis GEN3013 and its effects on healthy body cells.

In the introduction, the authors should clearly state the purpose of the work, and not just describe its results.

The introduction lacks brief information about the chemotherapeutics used in the work 

Author Response

However, the authors could point out that the introduction into the clinic must be preceded by further detailed studies of L. lactis GEN3013 and its effects on healthy body cells.

(Answer)

   Thank you so much for your kind comment. I understood and agreed with your concerns. So, following your advice, we validated the direct-killing effects of L. lactis GEN3013 in a normal cell line. Because the normal breast, colon, and lung cells could not be obtained immediately, we used the C2C12 cell line instead. The C2C12 is a mouse myoblast cell line that differentiates rapidly, forming contractile myotubes and producing characteristic muscle proteins.

   We seeded 2x104 C2C12 cells in a 96-well cell culture plate, and then treated with L. lactis GEN3013 lysates. Similar to the previous experiment, lysates were treated with 0.5% of the total volume and additionally, 5% lysates were added to confirm whether direct-killing effects of L. lactis GEN3013 were induced by bacterial contamination. Based on the results, the PBS control and L. lactis GEN3013 lysate showed no difference in cell viability of C2C12 cells (Revision figure 1). Therefore, it was concluded that L. lactis GEN3013 is non-toxic to C2C12 cells.

   However, L. lactis GEN3013 cannot be concluded to be bio-safe for all normal cells and tissues. To develop L. lactis GEN3013 as new drugs, the same proof will be required in breast, colon, and lung cells, as Reviewer#1 said. As part of a future study, we will verify the biosafety of L. lactis GEN3013 in various normal cell types. Thank you very much for pointing out an important point.

In the introduction, the authors should clearly state the purpose of the work, and not just describe its results.

Thank you for sharing your good opinion. As your advice, before summarizing our results in the introduction, we have added a sentence that may indicate the purpose of this study.

Line 84-86: For these reasons, identifying effective microbiome with antitumor effects is an important step in developing novel anticancer modalities and enhancing conventional anti-cancer therapies.

The introduction lacks brief information about the chemotherapeutics used in the work.

(Answer)

   Thank you for mentioning an important point. As you mentioned, it seems to have been mainly written about immunotherapy-related information. Therefore, chemotherapy-related research was added to the introduction in order to help the better understanding of this study.

   Line 53-56: Studies suggest that cancer chemotherapy efficacies and outcomes are associated with the gut microbiome [1]. For instance, strain-specific Bifidobacterium breve could improve the efficacy of oxaliplatin in the syngeneic mice model [2], and targeting F. nucleatum in the tumor augments the effect of chemotherapy against colorectal cancers [3].

Reviewer 3 Report

I find the article well written, easy to read and follow. The science is sound and it is of importance in the field and could be helpful in decision making in the healthcare.
There are no major issues with the manuscript. I'm only curious about the some background rationals.

1. How the treatment concentrations were determined, especially in mice and co-treatments. Is there any pilot experiment, or reference?

2. Among many chemotherapy medications, why the authors reported the combination use of L. lactis GEN3013 with Oxaliplatin and Pemetrexed. Please consider to add if there were hypothesis in mechanic of actions, or any negative results with other medications demonstrating the screening process.

Author Response

   Thank you very much for your positive feedback. We agree that there was a lack of some background and information. Therefore, additional explanations are provided below.

1. How the treatment concentrations were determined, especially in mice and co-treatments. Is there any pilot experiment, or reference?

(Answer)

   The first step of determining the treatment concentration of bacteria in the syngeneic mice model was based on the references. Most bacteria administration and colonization experiments used 1-5x109 CFU per mouse [1-3]. Based on references, we conducted a pilot experiment with 1x108, 1x109, and 1x1010 CFU. It was confirmed that 1x109 CFU was sufficient to exert an antitumor effect although data are not shown. 

   Especially in the case of co-treatment, if the concentration of an anticancer agent or bacteria is too high, one effect will dominate so the combination effect cannot be confirmed. Therefore, it is necessary to select a suitable concentration that can exhibit a combination effect while having a single effect. Therefore, experiments were conducted with several doses of bacteria, 1x108, 1x109, and 1x1010 CFU, and all concentrations were suitable to show the combination effect while the a-PD-1 effect was maintained (Figure 4E). As a result, all bacteria experiments in mice were conducted with 109 CFU per mouse, which is the optimal concentration that shows both the single effect and the combination effect.

2. Among many chemotherapy medications, why the authors reported the combination use of L. lactis GEN3013 with Oxaliplatin and Pemetrexed. Please consider to add if there were hypothesis in mechanic of actions, or any negative results with other medications demonstrating the screening process.

(Answer)

   Most chemotherapy is known to suppress or weaken the immune system [4,5] in addition to killing cancer cells. For this reason, some chemotherapy drugs are also used as immunosuppressants for severe autoimmune diseases [6]. For example, methotrexate [7] and cyclophosphamide [8] are commonly used for autoimmune diseases because they could impair the proliferative and/or effector functions of peripheral T cells. However, recent studies revealed that oxaliplatin and pemetrexed could induce immune-boosting effects by facilitating tumor-infiltration of T cells and NK cells and increasing T cell activation [9-12]. L. lactis GEN3013 also showed immune-stimulating ability in this study (Figure 5). Comprehensively, the immune-boosting ability of L. lactis GEN3013 was expected to improve the efficacy of oxaliplatin and pemetrexed by giving synergistic effects in terms of its immunogenic activity. 

   Although specific chemotherapeutic agents were selected based on the references in this study, identifying synergistic and non-synergistic agents with L. lactis GEN3013 would be very useful. Next, we will investigate the relationship between each agent's mechanism of action and the combined effect with L. lactis GEN3013. Your great advice has allowed us to proceed with further study in a more meaningful way. Thank you again.

Round 2

Reviewer 1 Report

The authors provide satisfactory responses and have addressed the concerns appropriately.